# Engineering of a bona fide light-operated calcium channel

Lian He[1,6], Liuqing Wang [2,6], Hongxiang Zeng[3], Peng Tan[1], Guolin Ma [1], Sisi Zheng [2], Yaxin Li[2], Lin Sun[2], Fei Dou [4], Stefan Siwko[1], Yun Huang [3,5✉], Youjun Wang [2✉] & Yubin Zhou [1,5✉]

The current optogenetic toolkit lacks a robust single-component $Ca^{2+}$-selective ion channel tailored for remote control of $Ca^{2+}$ signaling in mammals. Existing tools are either derived from engineered channelrhodopsin variants without strict $Ca^{2+}$ selectivity or based on the stromal interaction molecule 1 (STIM1) that might crosstalk with other targets. Here, we describe the design of a light-operated $Ca^{2+}$ channel (designated LOCa) by inserting a plant-derived photosensory module into the intracellular loop of an engineered ORAI1 channel. LOCa displays biophysical features reminiscent of the ORAI1 channel, which enables precise optical control over $Ca^{2+}$ signals and hallmark $Ca^{2+}$-dependent physiological responses. Furthermore, we demonstrate the use of LOCa to modulate aberrant hematopoietic stem cell self-renewal, transcriptional programming, cell suicide, as well as neurodegeneration in a *Drosophila* model of amyloidosis.

[1] Center for Translational Cancer Research, Institute of Biosciences and Technology, Texas A&M University, Houston, TX, USA. [2] Beijing Key Laboratory of Gene Resource and Molecular Development, College of Life Sciences, Beijing Normal University, Beijing, China. [3] Center for Epigenetics and Disease Prevention, Institute of Biosciences and Technology, Texas A&M University, Houston, TX, USA. [4] Beijing Key Laboratory of Genetic Engineering Drugs and Biotechnology, College of Life Sciences, Beijing Normal University, Beijing, China. [5] Department of Translational Medical Sciences, College of Medicine, Texas A&M University, Houston, TX, USA. [6] These authors contributed equally: Lian He, Liuqing Wang. ✉email: yun.huang@tamu.edu; wyoujun@bnu.edu.cn; yubinzhou@tamu.edu

The prototypical $Ca^{2+}$ release-activated $Ca^{2+}$ (CRAC) channel, composed of ORAI1 and the stromal interaction molecule 1 (STIM1), constitutes an important $Ca^{2+}$ entry route in mammalian cells[1–4]. Aberrant STIM1–ORAI1 signaling has been intimately linked to immunoinflammatory disorders, myopathy, tumorigenesis, and neurodegenerative diseases, making the CRAC channel a potential therapeutic target[5–7]. Recently, plant-derived photosensitive modules have been engineered into STIM1 to generate genetically encoded $Ca^{2+}$ actuators (GECA)[5,8–11], thereby enabling remote and noninvasive control of CRAC channel-mediated $Ca^{2+}$ entry into cells[12–15]. However, STIM1-based GECAs have two intrinsic drawbacks: the absolute requirement of endogenous ORAI channels and the potential side effects arising from crosstalk with other STIM1-associated targets, such as transient receptor potential (TRP) channels and voltage-gated $Ca^{2+}$ ($Ca_V$) channels[16–19]. Other optogenetic tools capable of inducing transient intracellular $Ca^{2+}$ mobilization include light-activated chimeric G protein-coupled receptors (GPCRs) and receptor tyrosine kinases[20–23]. These engineered receptors, nonetheless, could lead to the co-activation of diacylglycerol (DAG)-mediated signaling to initiate non-$Ca^{2+}$-related physiological responses. Furthermore, attempts have been made to generate channelrhodopsin-2 (ChR2) variants with increased selectivity for $Ca^{2+}$ over other cations[24], but they generally fail to match the exceptional $Ca^{2+}$ selectivity seen in CRAC channels ($P_{Ca}/P_{Na} > 1000$)[2].

To overcome these hurdles, we set out to engineer a single-component light-operated $Ca^{2+}$ (LOCa) channel by inserting the light-oxygen-voltage domain (LOV2) of *Avena sativa* phototropin 1 into various regions of ORAI1[25], a four-pass transmembrane (TM) protein that constitutes the pore-forming subunit of the CRAC channel[1–4]. We envision that photon-induced conformational changes in LOV2 could trigger allosteric ORAI1 gating (Fig. 1a). Rational design, randomized mutations via error-prone PCR, and high-throughput fluorescence-based screening assays were employed to generate LOCa as a light-gated $Ca^{2+}$ channel that could reversibly mediate $Ca^{2+}$ influx without the need for exogenous cofactors.

## Results

**Design and optimization of LOCa.** Inspired by successful design of a light-activatable viral potassium channel designated BLINK1[26], we set out to engineer a more challenging target, the ORAI1 channel made of four-pass TM domains and assembled as a hexamer in the plasma membrane[27,28]. LOV2 was initially inserted into different regions known to be critical for ORAI1 channel activation[29,30], including the N/C termini and various loop regions (Supplementary Fig. 1a). The resulting LOCa variants were tested with high-throughput $Ca^{2+}$ imaging, using GCaMP6m fluorescence as readout, for their ability to mediate blue light-induced $Ca^{2+}$ influx. However, LOV2 fusion or insertion at these regions invariably failed to evoke light-induced intracellular $Ca^{2+}$ changes (Supplementary Fig. 1a). It is likely that the free energy generated by the LOV2 conformational switch (~3.8 kcal/mol) is insufficient to shift ORAI1 $Ca^{2+}$ channel from a closed state to an open configuration[25,28,31]. We therefore reasoned that using constitutively active ORAI1 (caORAI1) mutants as the engineering template might reduce the energy cost to enable allosteric gating of the ORAI1 channel with light. After testing four reported caORAI1 mutants[32–34] with LOV2 inserted in the intracellular loop connecting TM2 and TM3, we discovered that the ORAI1(P245T)-LOV2 hybrid construct (named as LOCa1, with LOV2 insertion between residues R167 and M168) showed the largest light-induced $Ca^{2+}$ influx ($F_{max}/F_0$: ~1.5; Fig. 1b). By optimizing the LOV2 insertion site within the intracellular loop (Fig. 1c,d), we further enhanced the

dynamic range of $Ca^{2+}$ changes from 1.5 to 2.1, with the best construct (LOV2 insertion at Site 6 between residues S163 and P164; Fig. 1c–e) named as LOCa2. Unexpectedly, LOCa2 exhibited a biphasic light-induced response of cytosolic $Ca^{2+}$: LOCa2-expressing cells showed an initial drop in the GCaMP6m fluorescence intensity within the first 12 s, followed by a 1.1-fold increase in the fluorescent signal with an activation half-life of 50 s (Fig. 1e). We further used the downstream $Ca^{2+}$-responsive transcription factor, the nuclear factor of activated-T cells (NFAT), as an independent readout for $Ca^{2+}$ signals. In HeLa cells expressing LOCa2, we found that ~90% of cells showed nuclear entry before light stimulation (Fig. 1f), implying that the engineered LOV2–ORAI1 hybrid channel could not be fully caged in the dark.

The biphasic $Ca^{2+}$ response and non-negligible basal activation of LOCa2 in the dark make it less ideal for precise control of cell signaling in biological systems, prompting us to carry out a third round of optimization. Because TM3–TM4 helix coupling is intimately involved in transducing STIM1-induced gating of the ORAI1 channel[33], we envisioned that introduction of additional mutation(s) into TM3 and the second extracellular loop might stabilize the dark state of engineered LOCa proteins (Fig. 1c). We therefore combined random mutagenesis using error-prone PCR with high-throughput screening to evolve an improved hybrid channel (Fig. 1g). By using NFAT nuclear entry and $Ca^{2+}$ influx as readouts, we identified one construct bearing the double mutations H171D/P245T (designated as LOCa3) that showed the least activation in the dark (Fig. 1h), as reflected by non-appreciable NFAT accumulation in the nuclei in the dark state (Fig. 1i). More importantly, LOCa3 exhibited a higher dynamic range of light-activatable $Ca^{2+}$ influx ($F_{max}/F_0$: ~3), with good reversibility ($t_{1/2,on}$ = 48.69 ± 4.53 s; $t_{1/2,off}$ = 56.84 ± 3.79 s; Fig. 2a) and faster activation kinetics compared to LOCa2 (time to plateau: 90 s vs 120 s; Supplementary Fig. 1b, c). Importantly, LOCa3 also enabled precise spatial control of $Ca^{2+}$ signaling, as reflected in spatially confined generation of $Ca^{2+}$ signals when photo-illumination was sequentially applied to two neighboring cells (Fig. 2b and Supplementary Movie 1). The surface localization of LOCa3 was confirmed by immunocytochemical staining of FLAG-tagged LOCa3 in intact cells, with the FLAG peptide inserted in the second extracellular loop that is known to tolerate epitope insertion without compromising its light-induced effect (Supplementary Fig. 2a). To minimize the size of LOCa3, we further generated several truncated constructs by removing one or two portions of intracellular regions (Supplementary Fig. 2b). We found that deletion of the N-terminal 64 residues did not seem to compromise the light-gated function of LOCa3 (Supplementary Fig. 2b, c), making it a more compact optogenetic $Ca^{2+}$ generating tool. In the dark, LOCa3 exhibited minimal basal activity (Supplementary Fig. 3a). The degree of $Ca^{2+}$ influx could be tuned by varying the input light power densities (Supplementary Fig. 3b). To investigate the potential perturbation of LOCa3 on SOCE, we examined endogenous SOCE in HEK293 cells, with and without the co-expression of LOCa3. It turned out that HEK293 cells expressing LOCa3 showed slightly attenuated SOCE responses (Supplementary Fig. 3c), suggesting that LOCa3 might exert a limited dominant-negative effect on endogenous SOCE. The effects will also likely be variable and thus manageable, depending on the expression level of the engineered ORAI channel. Furthermore, we found that HEK293T cells depleted of ORAI (ORAI-KO) or STIM (STIM-KO) showed similar photo-induced $Ca^{2+}$ influx signals as the native unmodified cells (Supplementary Fig. 4a). Clearly, LOCa3 obviated the requirement of endogenous ORAI or STIM, making it an attractive tool to photo-manipulate tissues/cells with no or little ORAI expression. To validate this point, we

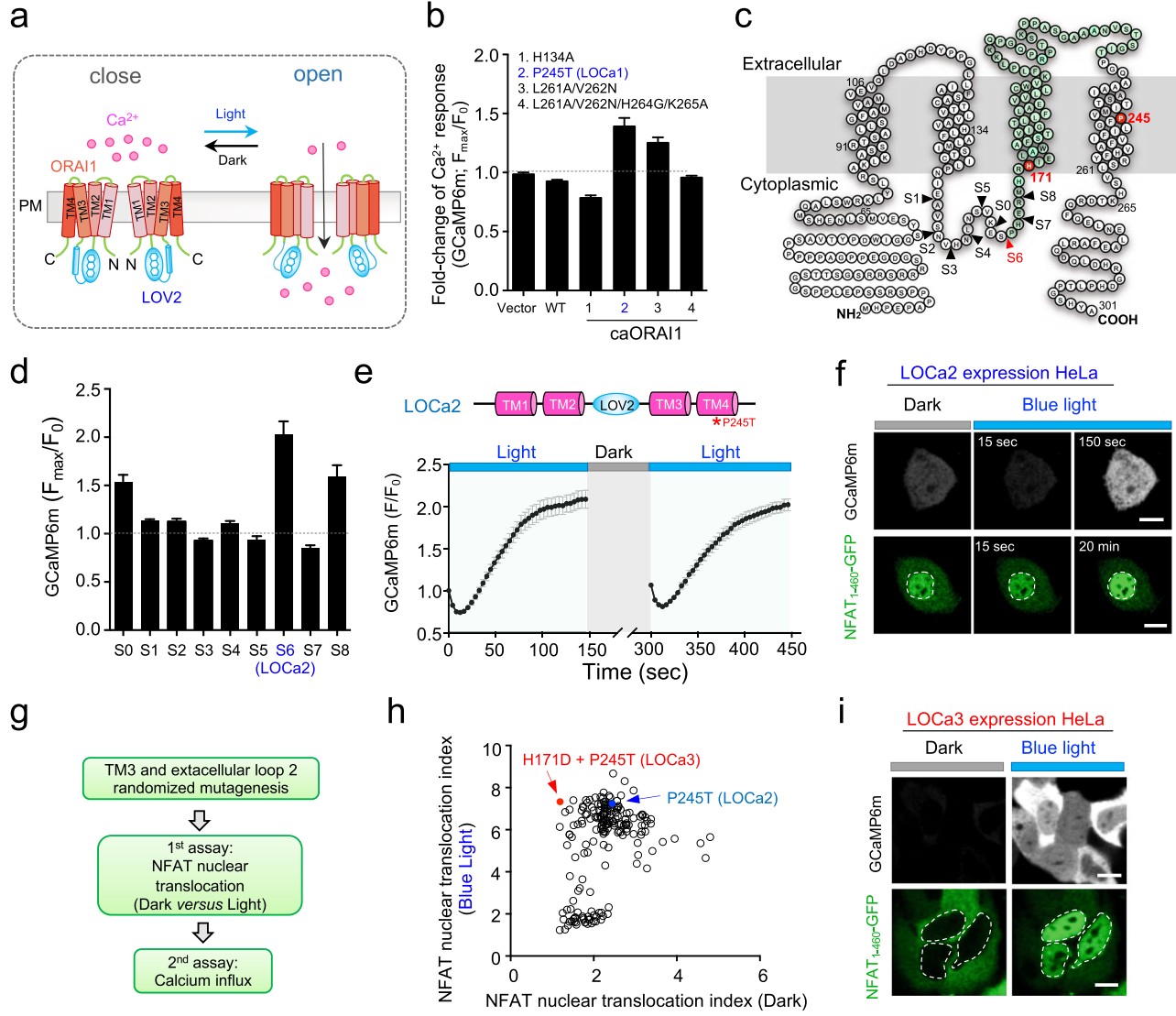

**Fig. 1 Design of a light-operated calcium channel (LOCa).** Data are shown as mean ± s.e.m. **a** Schematic depiction of photo-switchable Ca²⁺ influx through an engineered ORAI1 Ca²⁺ channel. The LOV2 domain is inserted into the intracellular loop of a constitutively active ORAI1 (caORAI1), which maintains the hybrid channel in a largely closed state in the dark. Upon photosimulation at 470 nm, conformational changes within LOV2 trigger allosteric activation of engineered ORAI1 to evoke Ca²⁺ flux across the plasma membrane. **b** The fold-change of photo-induced Ca²⁺ responses reported by GCaMP6m in HeLa cells expressing LOV2–ORAI1 hybrid variants. LOV2 was inserted between residues R167 and M168. The mutant P245T showed the most notable light-induced changes in intracellular Ca²⁺ (designated LOCa1). n = 24–52 cells. **c** Snake-like diagram of the ORAI1 Ca²⁺ channel. The LOV2 insertion sites tested in the study are indicated as arrowheads. TM3 and the second extracellular loop regions targeted for randomized mutagenesis are highlighted in green. **d** Comparison of light-induced Ca²⁺ changes after inserting LOV2 into the indicated positions of the ORAI1-P245T variant. S6 showed the highest light-dependent changes and was named as LOCa2. n = 22-66 cells from three independent assays. **e** Light-induced changes in cytosolic Ca²⁺ reported by GCaMP6m in HeLa cells transfected with LOCa2. Two cycles of light stimulation were applied. The domain architecture of the construct is shown above the curve. Blue bar, photo-illumination at 470 nm with a power density of 40 µW/mm². n = 24 cells from three independent assays. **f** Blue light modulated Ca²⁺ entry (top) and NFAT nuclear translocation (bottom) in HeLa cells expressing LOCa2. Scale bar, 10 µm. **g** The experimental flow for high-throughput screening of evolved LOCa constructs. **h** Biplot showing the degrees of NFAT nuclear translocation in HeLa cells expressing evolved mutants, either in the dark (x-axis) or under photo-illumination (y-axis). The construct showing the least dark activation and high Ca²⁺ response in the lit-condition was designated LOCa3, which bears two mutations in ORAI1: H171D and P245T. **i** Representative confocal images showing blue light-triggered Ca²⁺ response and changes in subcellular localization of NFAT₁₋₄₆₀-GFP. Scale bar, 10 µm.

compared the light-induced behavior of a STIM1-based optogenetic tool, Opto-CRAC[11,12,35], with LOCa3 side-by-side in different cell lines with varying ORAI1 expression levels (Supplementary Fig. 4b). Light-induced signals generated by Opto-CRAC[12] clearly exhibited a positive correlation with the expression levels of endogenous ORAI1 in the host cells. By contrast, LOCa3-induced Ca²⁺ signals remained largely unaffected by the varying degrees of ORAI1 expression in the same set

of cell lines (Supplementary Fig. 4b). As anticipated, the source of Ca²⁺ was found to be solely from the extracellular media, because removal of extracellular Ca²⁺ abrogated light-induced Ca²⁺ influx (Supplementary Fig. 4c). Lastly, we found that the expression of LOCa3 in mammalian cells did not seem to alter the basal Ca²⁺ levels, as reflected in the similar fluorescent signals of a ratiometric Ca²⁺ indicator (Supplementary Fig. 4d). Collectively, we have clearly demonstrated the use of LOCa3 to

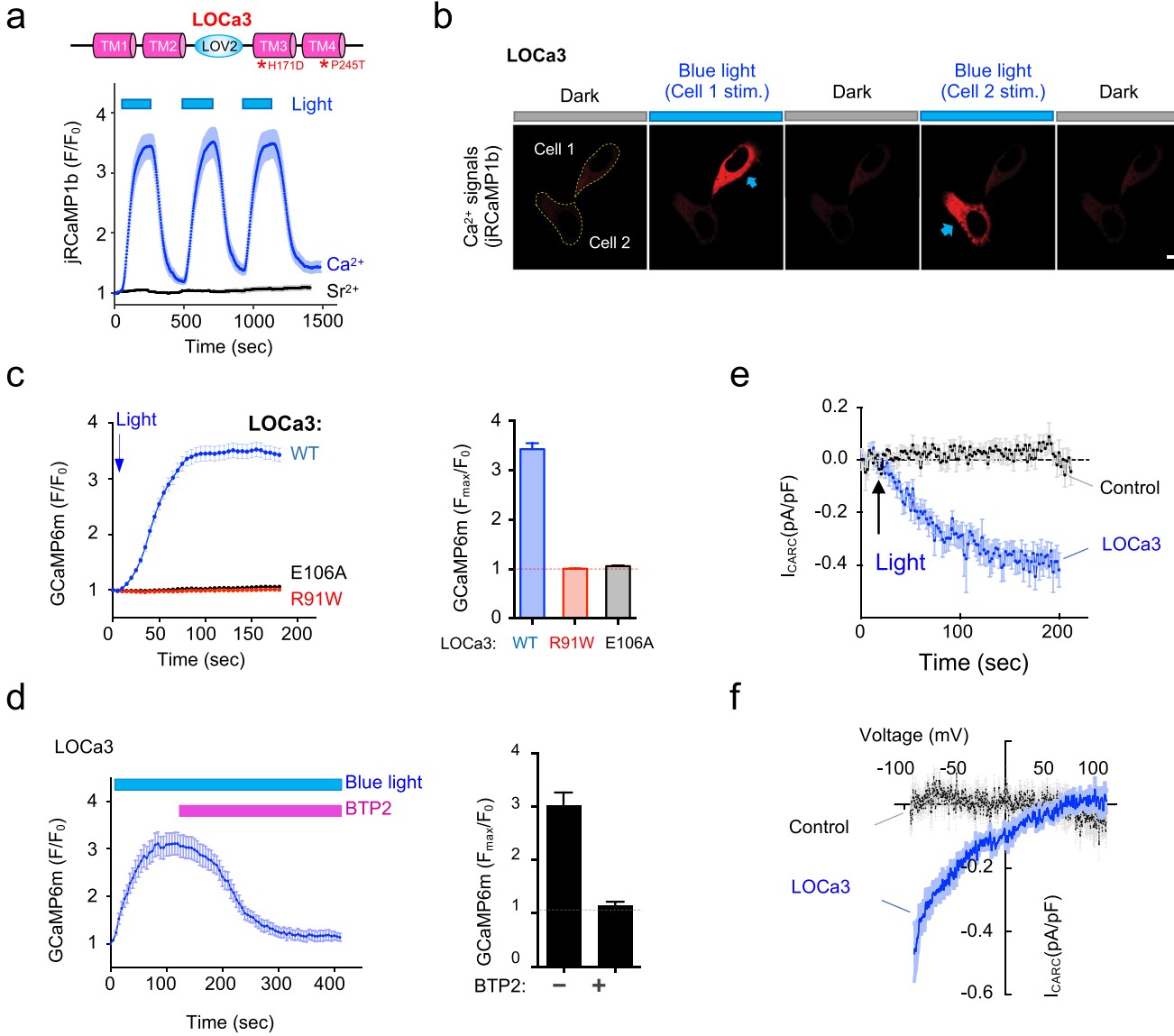

**Fig. 2 Characterization of LOCa3 in mammalian cells.** Data are shown as mean ± s.e.m. **a** A typical reversible influx of $Ca^{2+}$ (blue), but not $Sr^{2+}$ (black), reported by a red genetically encoded $Ca^{2+}$ indicator (jRCaMP1b) in ORAI-null HEK293 cells transfected with LOCa3. Cells were cultured in an imaging buffer containing 2 mM $Sr^{2+}$ or $Ca^{2+}$ and then subjected to three repeated light–dark cycles of stimulation. The activation and deactivation half-lives were 48.7 ± 4.5 s and 56.8 ± 3.8 s, respectively. ORAI1 channel is known to uniquely prohibit $Sr^{2+}$ influx because of its strict ion selectivity toward $Ca^{2+}$. $n = 14$-16 cells from three independent biological replicates. **b** Spatial control of $Ca^{2+}$ influx in two neighboring HeLa cells. Cells in the two indicated regions (dashed lines) were sequentially subjected to photo-illumination using a 488-nm laser (0.5% input). Scale bar, 10 μm. Also see Supplementary Movie 1. **c** Comparison of light-induced $Ca^{2+}$ influx among the indicated LOCa3 variants. The channel-inactivating mutations R91W and E106A completely blocked photo-triggered $Ca^{2+}$ entry. The quantification of the relative change of GCaMP6m signals after light stimulation is shown in the bar graph on the right. $n = 48$-52 cells from three biological replicates. **d**. BTP2 as a CRAC channel blocker effectively suppressed blue-light-induced $Ca^{2+}$ influx. Left, the time-course of GCaMP6 fluorescence. Right, quantification of the GCaMP6 signals before and after BTP2 treatment (5 μM). $n = 10$ cells. **e**. The mean time-courses of whole-cell currents in HEK cells expressing LOCa3 or a control vector following exposure to blue light illumination. $n = 8$-9 cells. **f**. Mean current-voltage relationships at the peak of light-induced currents ($n = 8$ cells).

achieve both temporal and spatial control of $Ca^{2+}$ signals, regardless of the presence of endogenous ORAI and STIM expression, in mammalian cells.

**Biophysical properties of LOCa3.** Next, we carried out further pharmacological and biophysical studies on LOCa3. To ensure that $Ca^{2+}$ ions indeed travel across the plasma membrane through the engineered ORAI1 channels, we introduced two inactivating mutations, the pore-dead E106A mutation and a dominant-negative mutation, R91W, that causes human severe combined immunodeficiency (SCID)[36–38], into LOCa3. Both

mutations abrogated light-induced $Ca^{2+}$ responses in transfected HEK293 cells (Fig. 2c), confirming that the engineered ORAI1-LOV2 hybrid protein indeed mediates photon-induced $Ca^{2+}$ influxes. Meanwhile, the CRAC channel inhibitor BTP2 abolished LOCa3 activity (Fig. 2d), indicating that the pharmacological properties of engineered LOCa3 remain similar to those of ORAI1. Furthermore, whole-cell patch clamping studies revealed that the light-induced $Ca^{2+}$ current through LOCa3 showed an inwardly rectifying I–V relationship (Fig. 2e, f) that is typical of the CRAC channel[2]. Furthermore, unlike $CaV_{1.2}$ channels that could mediate both $Ca^{2+}$ and $Sr^{2+}$ permeation across the PM

(Supplementary Fig. 5a, b), LOCa3 did not produce light-induced $Sr^{2+}$ influx while generating a robust light-induced $Ca^{2+}$ response (Fig. 2a), suggesting that the insertion of LOV2 into ORAI1 did not compromise its $Ca^{2+}$ selectivity. To further examine the $Ca^{2+}/Na^+$ selectivity of LOCa3, we carried out whole-cell patch clamp recordings on LOCa3-expressing cells bathed in an extracellular solution containing 0 $Ca^{2+}$ and 130 mM $Na^+$. We found that LOCa3 failed to mediate any discernible $Na^+$ influx current upon photostimulation (Supplementary Fig. 5c), and thus retained a high selectivity for $Ca^{2+}$ over $Na^+$. These findings imply that the engineered photo-switchable LOCa3 channel is biophysically similar to ORAI1. Taken together, these data establish LOCa3 as a light-gated $Ca^{2+}$ channel with high $Ca^{2+}$ selectivity reminiscent of the native ORAI1 $Ca^{2+}$ channel[2].

**Optical control of $Ca^{2+}$-modulated biological processes**. We next applied LOCa3 in cellulo and explored its potential to modulate $Ca^{2+}$-dependent cellular functions. Because $Ca^{2+}$ homeostasis has been closely implicated in the maintenance of hematopoietic stem cell (HSC) stemness[39,40], we asked whether optogenetic intervention by LOCa3 can be applied to regulate the cell fate of HSCs. To test this in a disease-relevant context, we used hematopoietic stem and progenitor cells (HSPCs) isolated from a mouse model with augmented HSC self-renewal and enhanced hematopoiesis upon disruption of the Ten-eleven Translocation 2 (*Tet2*) gene[41,42]. Given that pharmacological inhibition of $Ca^{2+}$ influx has been shown to enhance the maintenance of HSCs whereas augmented $Ca^{2+}$ influx causes loss of HSC self-renewal[43], we reasoned that LOCa3-mediated optogenetic activation of $Ca^{2+}$ signals could be exploited to suppress the abnormal self-renewal of *Tet2*-deficient HSPCs (Fig. 3a). We therefore transduced HSPCs isolated from wild-type (WT) and *Tet2* knockout (*Tet2*-KO) mice with a retrovirus encoding LOCa3 (with co-expressed mCherry, mCh, as a marker). LOCa3-expressing mCh-positive cells displayed a marked increase of Fluo-4 signals upon light stimulation, confirming the photon-triggered $Ca^{2+}$ influx in HSPCs (Fig. 3b, c). We next determined the in vitro self-renewal capacity of both WT and *Tet2*-KO HSPCs, in the absence or presence of light illumination, by analyzing the frequency of the Lin-negative (Lin−), c-Kit+ Sca-1+ population (LSK) with flow cytometry (Fig. 3d and Supplementary Fig. 6). Consistent with previous reports[41,42], *Tet2* ablation led to a marked increase of the LSK population under normal non-lit conditions (Fig. 3d). Upon light illumination, the *Tet2*-deficient group displayed a 15% reduction of LSK cells (Fig. 3d), indicating that light-induced $Ca^{2+}$ entry indeed suppressed HSPC self-renewal. Under the same treatment, normal HSPCs only showed an appreciable (~2%) but non-significant ($P = 0.13$) reduction in the LSK pool (Fig. 3d, e). These results suggest that optogenetic perturbation of $Ca^{2+}$ signaling in the hematopoietic system might hold promise to curtail the aberrant self-renewal of HSCs bearing TET2 loss-of-function mutations, which are frequently detected in clonal hematopoiesis and various hematological malignancies[44,45].

We then moved on to examine whether light-induced $Ca^{2+}$ influx can be applied to precisely control gene expression. To report $Ca^{2+}$-dependent transcriptional activity, we used a synthetic construct containing a luciferase reporter gene *luc2P* (*Photinus pyralis*) under the control of a minimal promoter with a $Ca^{2+}$-responsive NFAT response element (NFAT-RE; Fig. 3f). Upon blue light stimulation, we observed a light-tunable increase of bioluminescence in LOCa3-expressing cells but not in the control group with the pore-dead ORAI1 mutant E106A (Fig. 3g). Next, we replaced the luciferase reporter with an N-terminal

fragment of mixed lineage kinase domain-like protein (MLKL-N; residues 1-190) that constitutively induces cell death via necroptosis[46]. In the light-treated group, we observed an increase of cell death as indicated by SYTOX blue staining, which turned blue when entering dying cells with a compromised PM but remained impermeant and thus non-fluorescent for intact cells (Fig. 3h). Collectively, these findings have confirmed the successful design of an optogenetic suicide device based on LOCa3.

To achieve transcriptional control over endogenous genomic loci, we combined LOCa3 with CaRROT ($Ca^{2+}$-responsive transcriptional reprogramming tool)[35] and examined the light-induced changes in the transcription of endogenous *MYOD1* (myogenic differentiation 1) (Fig. 3i). CaRROT was previously developed by us to rewire $Ca^{2+}$ signals for transcriptional programming, which contains three key elements, a catalytically dead dCas9, the VP64 transcriptional coactivator, and the N-terminal non-DNA-binding regulatory domain of NFAT (NFAT$_{reg}$) that enables nucleocytoplasmic shuttling in response to $Ca^{2+}$ fluctuations in the cytosol[35]. In the presence of sgRNA targeted to the promoter region of *MYOD1*, we observed a robust light-dependent induction of target gene expression (Fig. 3j), thereby establishing the feasibility of using LOCa3 for light-switchable transcriptional programming in mammalian cells.

**Applying LOCa3 to alleviate neurodegeneration in vivo with light**. Finally, we tested the feasibility of LOCa3 for in vivo optogenetic intervention. Impaired store-operated $Ca^{2+}$ entry (SOCE) mediated by STIM-ORAI signaling has been implicated in the pathogenesis of neurodegenerative disorders such as Alzheimer's disease (AD)[47,48]. Since boosting SOCE activity via pharmacological or genetic manipulations could ameliorate AD syndromes, we envision that optogenetic stimulation of neuronal calcium signaling might rescue the impaired neuronal SOCE activity, which has been linked to neurodegenerative disease in both animal models and patients[47,48]. To rapidly test our hypothesis in vivo, we resorted to a *Drosophila* model of amyloidosis by pan-neuronal expression of the AD-linked human amyloid beta 42 (Aβ$_{42}$) gene, a model that is commonly used to study age-dependent neurodegeneration in a living organism[49]. Neuron-specific expression of Aβ$_{42}$ and/or LOCa3 was achieved by crossing the driver strain (bearing the GAL4 module under the control of the neuronal *elav* promoter) with the effector strains, in which the target gene expression was conditionally activated by the GAL4-UAS system (Fig. 4a). We used transgenic flies co-expressing an improved red, genetically encoded $Ca^{2+}$ indicator, jRCaMP1b[50], to monitor $Ca^{2+}$ changes within *Drosophila* cells in vivo. Following blue light stimulation, we observed a significant increase of red fluorescent signals in the fly brain by 12% on average (Fig. 4b). By contrast, the control group (Aβ42-expressing flies without LOCa3) did not exhibit light-induced changes in the jRCaMP1b fluorescence (Fig. 4c). These results validated functional LOCa3 expression to photo-trigger $Ca^{2+}$ influx in vivo.

To quantitatively assess neurodegenerative phenotypes with and without light stimulation, we used a climbing assay based on the startle-induced negative geotaxis response. Over the course of two months, flies progressively lost their climbing ability due to the normal aging process (left panel, Fig. 4d and Supplementary Movie 2). By contrast, in Aβ$_{42}$-expressing flies, we observed an accelerated progressive loss of climbing ability (middle panel, Fig. 4d). Within a short period of one month, nearly all the transgenic flies totally lost their climbing ability regardless of blue light stimulation (middle panel, Fig. 4d and Supplementary Movie 3). Importantly, in the experimental group expressing both Aβ$_{42}$ and LOCa3, we noted a light-dependent rescue of the

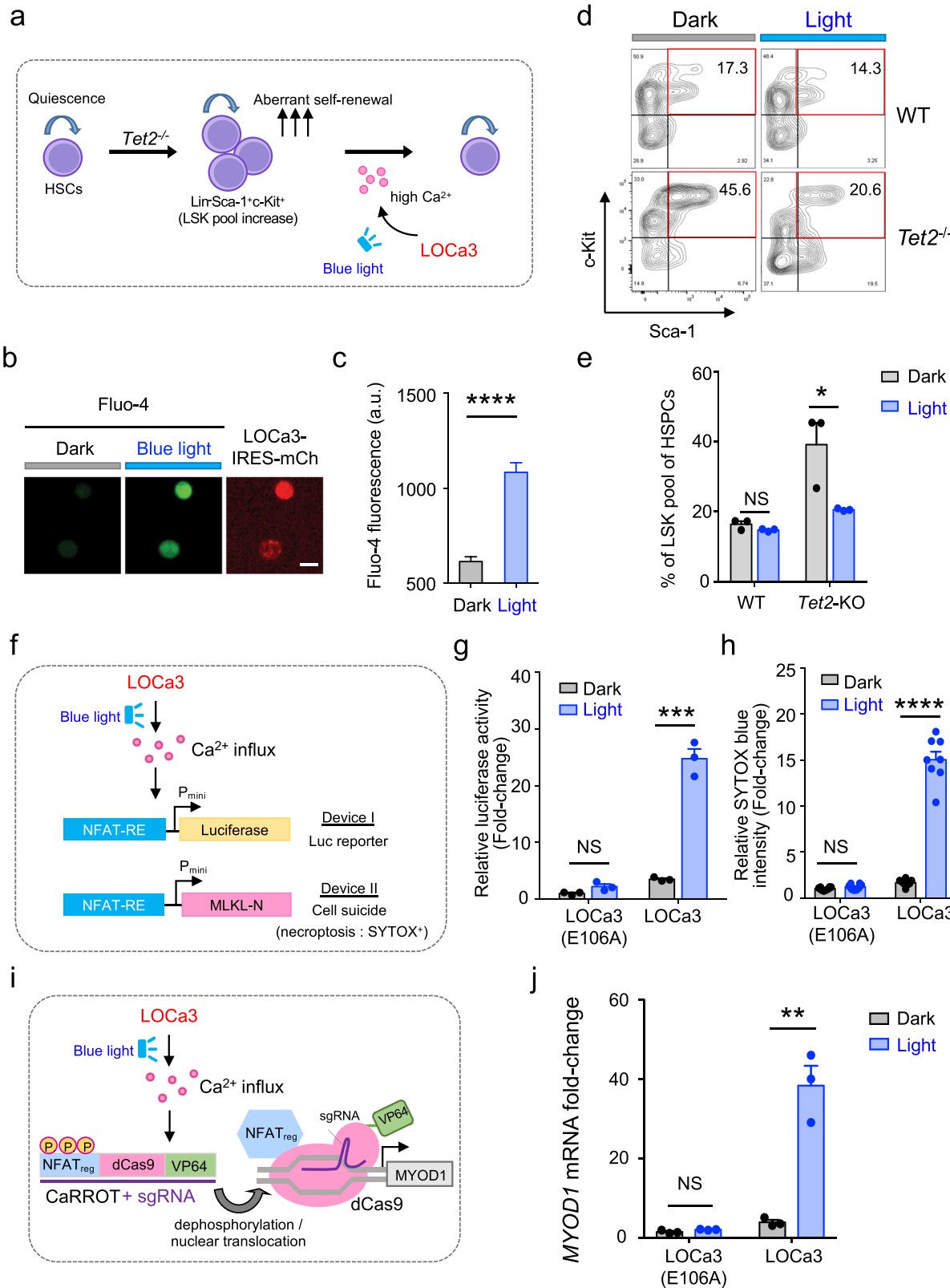

neurodegenerative phenotype. For transgenic flies subjected to blue light illumination (pulsed LED at 470 nm with a density of 40 μW/mm², 10 min for each hour per day), aged flies showed a significant improvement in their climbing ability (blue vs black traces; right panel, Fig. 4d and Supplementary Movie 4). Taken together, we have demonstrated the feasibility of optogenetic modulation of Ca²⁺ signals to intervene in neurodegeneration in vivo.

## Discussion

By engineering the LOV2 photosensitive domain into the intra-cellular loop of a constitutively active ORAI1 Ca²⁺ channel, we have constructed a series of blue light-gated Ca²⁺ channels termed LOCa with a small size (less than 1.5 kb), making it sui-table for most viral packaging systems. The pharmacological and biophysical features of LOCa resemble those of the native ORAI1

**Fig. 3 LOCa3 enables light-controllable regulation of biological processes.** Data are shown as mean ± s.e.m. Blue light was delivered using pulsed LEDs emitting at 470 nm (40 μW/mm$^2$). Two-tailed unpaired Student's $t$-test was used for the statistical test. **a** Diagram showing the use of LOCa3 to curtail aberrant increase in self-renewal of $Tet2^{-/-}$ HSCs. Enhanced $Ca^{2+}$ influx into HSPCs leads to loss of self-renewal due to perturbation of the $Ca^{2+}$-mitochondria axis to drive stem cell differentiation. Optogenetic activation of $Ca^{2+}$ entry can thus suppress the aberrant self-renewal of $Tet2$-null HSPCs. **b** Representative confocal images of Fluo-4 signals in LOCa3-expressing $Tet2^{-/-}$ HSPCs before and after photostimulation. HSPCs were infected with a retrovirus packaged with LOCa3-IRES-mCherry and loaded with Fluo-4 for $Ca^{2+}$ imaging. Scale bar, 10 μm. **c** Quantification of Fluo-4 signals in LOCa3-expressing $Tet2^{-/-}$ HSPCs before and after light illumination. $n = 151$ cells. $****P < 0.0001$. **d** Flow cytometry analysis of the self-renewal ability of normal (WT) or $Tet2^{-/-}$ HSPCs expressing LOCa3-IRES-mCh before and after photostimulation. Self-renewal ability was gauged by the frequency of the LSK cell population marked by c-kit$^+$Sca-1$^+$ staining. **e** The frequency of LSK pools (as shown in panel d) in WT or $Tet2^{-/-}$ HSPCs under the indicated conditions. $*P = 0.04$ ($n = 3$ independent biological replicates). $P = 0.1550$ for WT group. **f** Design of two synthetic gene expression devices based on LOCa3. Expression of the target genes was under the control of the $Ca^{2+}$-responsive NFAT response element (NFAT-RE). Positive nuclear staining for SYTOX blue indicated cell death. **g** Quantification of luciferase activity in HEK293 cells expressing LOCa3 in the absence (gray) or presence (blue) of photostimulation. $n = 3$ independent biological replicates. $***P = 0.003$. **h** Quantification of SYTOX blue staining of the indicated groups with and without photostimulation. $n = 8$ fields of view. $****P < 0.0001$. **i** Combining LOCa3 with CaRROT to enable optogenetic control over the transcription of endogenous genes. **j** Quantification of the mRNA levels of $MYOD1$ in HEK293 cells transfected with LOC3a and CaRROT/sgRNA before and after photostimulation. $n = 3$ independent biological replicates. $**P = 0.0023$.

channel, which is among the most $Ca^{2+}$-selective ion channels. Upon photostimulation, LOCa can be used to generate user-defined spatial and temporal patterns of intracellular $Ca^{2+}$ signals to deliver hallmark $Ca^{2+}$-modulated physiological responses. Compared with other optogenetic $Ca^{2+}$-modulation tools (Supplementary Table 1), the LOCa3-based tool has a relatively low basal activity, high $Ca^{2+}$ selectivity, no dependence on STIM or ORAI expression, and unique kinetic features. The activation half-life of LOCa3 is slower than STIM1-based tools, but the deactivation half-life lies between those of CRY2-based and LOV2-based GECAs. These properties make LOCa3 an ideal tool to control physiological processes with slower requirements on kinetics, such as gene expression, immunomodulation, and cell metabolism. When tested in stem cells, ectopic expression of LOCa3 could effectively suppress the aberrant self-renewal of defective HSPCs. LOCa3 can be further applied to enable light-inducible transcriptional programming and cell suicide, and can thus be exploited as a safety switch for adoptive cell therapies. Most importantly, we have demonstrated the use of LOCa3 to noninvasively intervene in neurodegeneration in a *Drosophila* model of AD. Being the leading cause of dementia in aged populations, AD imposes an escalating global burden on the health care system[51]. Optogenetic approaches described herein might hold promise for personalized neuromodulation to aid the future management of neurodegenerative disorders. Taken together, LOCa is a single-component, photo-switchable $Ca^{2+}$-selective channel that is amenable for many biotechnological and biomedical applications.

## Methods

**Plasmid construction**. Plasmids were created through standard restriction enzyme digestion-ligation and the NEBuilder HiFi DNA assembly methods. KOD Start DNA polymerase (EMD Millipore, MA, USA) was used for PCR amplification. The QuikChange Lightning Multi Site-Directed Mutagenesis Kit (Agilent Technologies) was used to introduce mutations. All other molecular cloning reagents were purchased from New England Biolabs (Ipswich, MA, USA) unless otherwise mentioned. All vectors were confirmed by Sanger DNA sequencing.

For LOV2–ORAI1 hybrid constructs, cDNA sequences encoding human ORAI1 (hORAI1) were PCR amplified and subcloned into a modified pcDNA3.1 (+) vector with mKate2-P2A pre-inserted between the NheI and BamHI sites. The amplified sequence encoding hORAI1 was digested with BglII and XhoI, and inserted into the pcDNA3.1(+)-mKate2-P2A vector following treatment with BamHI and XhoI. Activating ORAI1 mutations were individually introduced into pcDNA3.1(+)-mKate2-P2A-ORAI1 with the QuickChange Lightning Multi Site-Directed Mutagenesis Kit (Agilent Technologies). The AsLOV2$_{404-546}$ fragment flanked with BamHI and BspEI sites was inserted into selected regions of ORAI1 through NEBuilder® HiFi DNA Assembly (NEBuilder® HiFi DNA Assembly Master Mix, New England Biolabs Inc.). ORAI1 fragments (with LOV2 insertion) from LOCa3 were cloned into pcDNA3.1(+), pcDNA3.1(+)-mCh, pcDNA3.1 (+)-YFP or pmCherry-N1 vectors to make LOCa3(1–301), mCh-LOCa3 (1–301), mCh-LOCa3 (65–301), mCh-LOCa3 (65–286), YFP-LOCa3 (1–301), and LOCa3

(1–301)-mCh. To make FLAG-tagged LOCa3 (pcDNA3.1-LOCa3-FLAG), oligos encoding the FLAG epitope (DYKDDDDK) were inserted into the second extracellular loop of LOCa3 (as indicated in Supplementary Fig. 2a) through a standard PCR method.

To enable retroviral expression of LOCa3, cDNA encoding LOCa3 was cloned into a customized MSCV-IRES-mCherry vector between the EcoRI and NotI sites to yield MSCV-LOCa3-IRES-mCherry. The lentiviral expression vector of LOCa3 was made via the assembly of mCh-LOCa3 (1–301) gene fragment with the pLenti-puro backbone. The NFAT-dependent luciferase reporter plasmid pGL4.30[luc2P/NFAT-RE/Hygro] was purchased from Promega (Madison, WI, USA). To create the NFAT-dependent suicide plasmid, human MLKL-NT (1–190) was cloned into the same vector using NEBuilder® HiFi DNA Assembly to replace the luciferase reporter gene. The CaRROT system from a previous study[35] was used here, which contains NFAT-dCas9-VP64 and sgRNA (targeting $MYOD1$). For the fly transgenic expression vector, the LOCa3 fragment or jRCaMP1b was cloned into pValium20 between the XbaI and EcoRI sites. See Supplementary Table 2 for all the primers used in this study.

**Time-lapse confocal imaging and data processing**. To test light-induced $Ca^{2+}$ influx for LOCa variants, HeLa cells stably expressing GCaMP6m were seeded in 35 mm glass-bottom dishes (D35C4-20-1.5-N, Cellvis, Mountain View, CA, USA) and then transfected with individual constructs 20–24 h before imaging. Confocal imaging was performed on a Nikon Eclipse A1R microscope mounted onto a Nikon Eclipse T1 body. An incubation cage was installed to maintain the temperature, humidity, and $CO_2$ supplies. Lipofectamine 3000 (Thermo Fisher Scientific, MA, USA) was used for transfection following the manufacturer's protocol. Images for GCaMP6m and mKate2 (with 40x oil objectives) were acquired every 5 s for 2–3 min depending on the experimental requirements. The 488 nm laser was used for photoactivation with 5% output and pre-programmed dark–light cycles. BTP2 purchased from Sigma-Aldrich (# 203890, St. Louis, MO, USA) was used to test the CRAC-channel blocking activity at a concentration of 5 μM. For the NFAT nuclear translocation assay, plasmids encoding mKate2-P2A-LOCa variants were transfected into HeLa cells stably expressing NFAT$_{1-460}$-GFP. Images were acquired every 1 min for 15 min under blue light stimulation (470 nm, 40 μW/mm$^2$, ThorLabs, Inc.). Only the mKate2-positive cells (an indicator of expression) were used for analysis. The YFP fused LOCa3 was transfected into WT, STIM-KO (STIM1/STIM2 double KO), or ORAI-KO (ORAI1, ORAI2, and ORAI3 triple KO) HEK293T cells. The $Ca^{2+}$ changes were reported by jRCaMP1b and blue light stimulation was indicated in the figure. Similar experiments were performed when cells were bathed in 0 mM $Ca^{2+}$ extracellular solution. To monitor the resting cytosolic $Ca^{2+}$ levels of LOCa3 cells, a ratio-metric $Ca^{2+}$ indicator (GEM-GECO) was co-transfected with YFP-LOCa3 into HEK293T cells. Fluoresce intensity was recorded for BFP and Green channels without blue light stimulation. To measure the basal $Ca^{2+}$ level caused by LOCa3, we used either HEK 293 cells expressing a ratiometric $Ca^{2+}$ indicator, GEM-GECO, together with LOCa3 or an empty vector, or HeLa cells transfected with mKate2-P2A-LOCa3 were examined with Fluo-4 $Ca^{2+}$ (Thermo Fisher Scientific, F14201, Waltham, MA, USA) according to the manufacturer's protocol before imaging. GEM-GECO ratios between LOCa3-positive and control cells were compared. Fluo-4 fluorescence intensities were measured for both mKate2-positive and mKate2-negative cells (as control) under the dark state and after blue light illumination conditions. To detect the degrees of $Ca^{2+}$ entry of LOCa3 under different blue light power stimulation, HeLa cells co-transfected with mKate2-P2A-LOCa3 and GCaMP6m were used for the experiment. The average intensity of GCaMP6m for mKate2-positive cells (without blue light treatment) were calculated as background control. Other duplicated transfected groups were treated with light at varying power densities (as indicated in the figure) for 10 s, with the images immediately acquired using the GFP channel (with

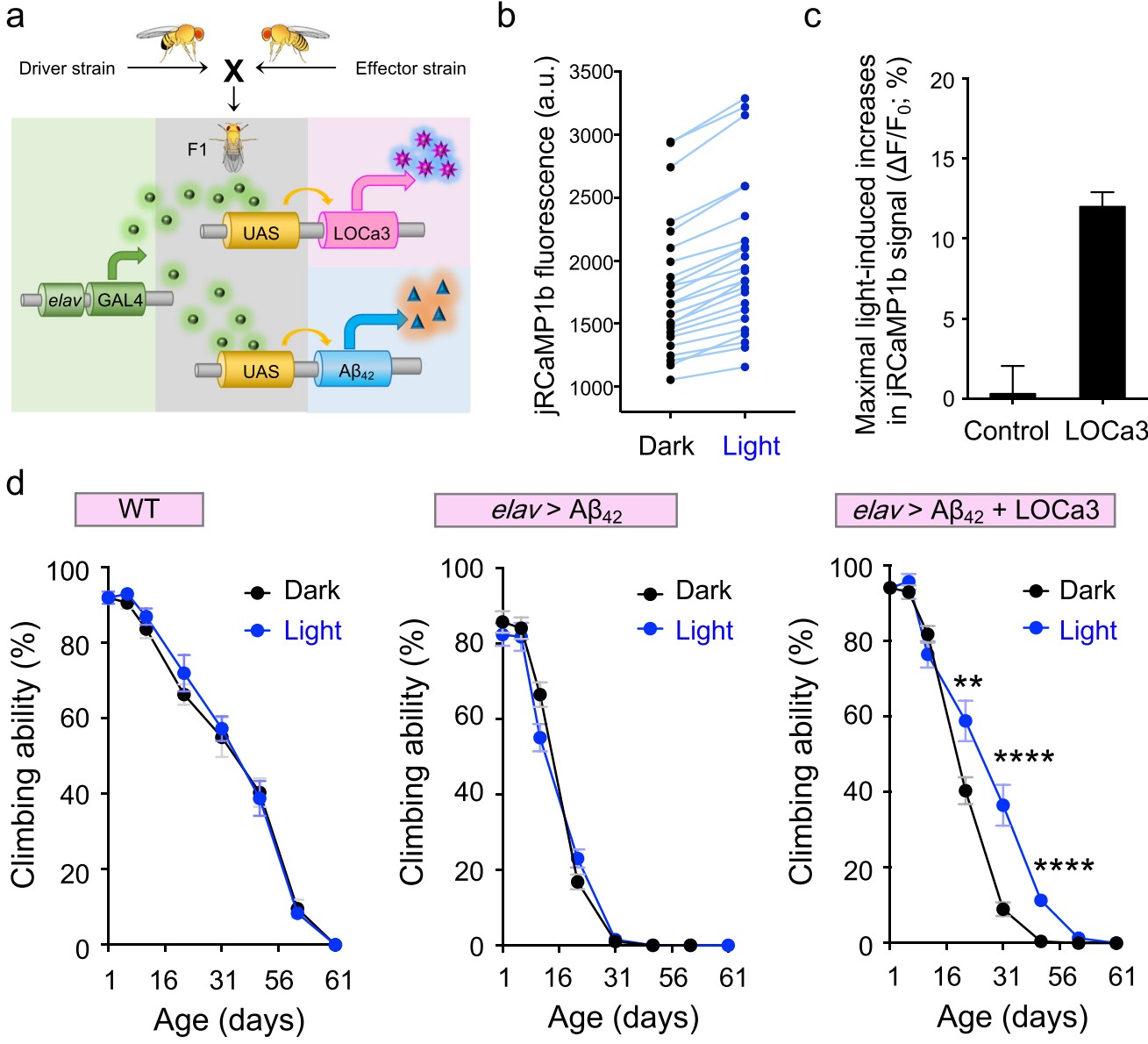

**Fig. 4 LOCa3 enables optogenetic intervention in neurodegeneration in vivo.** Data are shown as mean ± s.e.m. Blue light was delivered using pulsed LEDs emitting at 470 nm with a power density of 40 μW/mm². Flies were subjected to a total of 10 min photostimulation per hour for up to two months. **a** Diagram showing the generation of AD flies expressing Aβ₄₂ and /or LOCa3 by using the GAL4-UAS expression system, with the driver stain bearing GAL4 under the control of a *Drosophila* neuron-specific Elav (embryonic lethal abnormal visual system) promoter to enable pan-neuronal expression of target genes. **b** Statistics showing jRCaMP1b fluorescence within the same regions of the brain from LOCa3-expressing AD *Drosophila* before and after exposure to blue light ($n = 24$ from 5 flies). **c**. Quantification of changes of jRCaMP1b fluorescence before and after light stimulation in the brains of AD flies and AD flies co-expressing LOCa3 ($n = 5$ flies each). **d** Graphs showing the effects of photostimulation on the climbing ability during aging in WT (left panel), Aβ₄₂ (middle), and Aβ₄₂ + LOCa3 (right) flies (5 independent replicates; 10 files per repeat). **\*\*$P = 0.0046$; \*\*\*\*$P < 0.0001$** (compared to the dark group; two-tailed unpaired Student's $t$-test). Also see Supplementary Movies 2–4.

the same imaging setting). The fold-change over background control was plotted. MCF7, A549, U87, HeLa, HSkMC, and SH-SY5Y cells were transiently transfected or lentivirus transduced with mCh-Opto-CRAC or mCh-LOCa3, then cells were loaded with the Fluo-4 Ca²⁺ indicator as described above. Both the fluorescence intensity of Fluo-4 and mCherry were measured. To monitor light-induced Ca²⁺ influx in HSPCs, cultured cells were stained with Fluo-4. HSPC Ca²⁺ imaging data were acquired every 5 s for 5 min. To quantify GCaMP6m or Fluo-4 signals, we used the region-of-interest (ROI) toolbox in Nikon NIS-Elements software to define the cells. The "Time Measurement" tool was used to determine the fluorescent intensities for GCaMP6m or Fluo-4. The fluorescence intensity ratio ($F/F_0$ or $F_{max}/F_0$) was calculated and plotted as shown in the related figures.

**Generation of a LOCa library by error-prone PCR**. The ORAI1 fragment (residues 160–233) containing the distal half of the intracellular loop, TM3 and the second extracellular loop was subjected to randomized mutagenesis via error-prone

PCR. To achieve a high mutational rate, we chose to use the GeneMorph II Random Mutagenesis Kit (Agilent, Santa Clara, CA, USA). In brief, 10 ng of the template plasmid (mKate2-P2A-LOCa2) was used in a total reaction volume of 50 μl; 33 PCR cycles were used to increase the mutational frequency. The restriction site BspEI, which was introduced by LOV2 insertion at the C-terminus of LOV2, and an endogenous XmaI site in the second extracellular loop of ORAI1 were used for insertion of mutant fragments into the mKate2-P2A-LOCa2 vector. The PCR products flanked by BspEI and XmaI were sub-cloned into the digested vector to yield the LOCa variant library.

**High-throughput screening of LOCa variants**. HeLa cells stably expressing NFAT₁₋₄₆₀-GFP were seeded into 384-well glass-bottom dishes at a density of 2000 cells/25 μl medium/well by a multidrop dispenser (Thermo Fisher Scientific, Waltham, MA, USA). On the second day, the LOCa mutants obtained from error-prone PCR were individually transfected into seeded cells at a concentration of 25

ng/well by using Lipofectamine 3000. Four wells were repeated for each plasmid, with two repeats for the dark and lit conditions, respectively. Twenty-four hours after transfection, cells were either kept in the dark or subjected to blue light illumination with an external LED (470 nm, 40 μW/mm$^2$, ThorLabs, Inc.) for 20 min. Next, cells were fixed with 4% PFA and stained with DAPI following the standard fixing and nuclear staining protocol. All the staining and washing steps were performed by Hydrospeed (Tecan Diagnostics, Switzerland) and Multidrop Dispensers. The images for mKate2, GFP and DAPI were acquired by an IN Cell Analyzer 6000 (GE-Healthcare Life Sciences) with a 10x objective with 2 fields for each well. A total of 8 different imaging fields were acquired for each plasmid and each treatment. High-throughput imaging data were analyzed using Pipeline Pilot. The NFAT translocation index was used to indicate the degrees of nuclear translocation of NFAT as we used previously[52], with a high number representing a high level of nuclear translocation.

**NFAT-dependent luciferase expression.** For the NFAT-dependent luciferase expression assay, HEK293T cells were seeded in 96-well plates. LOCa3 and pGL4.30[luc2P/NFAT-RE/Hygro] (NFAT RE-luc, Promega, Madison, WI, USA) were co-transfected into cells. Sixteen hours after transfection, cells were treated with blue light (470 nm, 40 μW/mm$^2$, 1 min ON–9 min OFF cycles for 10 h) or kept in the dark as a control. Phorbol myristate acetate (PMA, Sigma-Aldrich, # P8139, St. Louis, MO, USA) was added to a final concentration of 15 nM. LOCa3 with the pore-dead mutation E106A was used as a control in the assay. Luciferase activity was determined by using a Bright-Glo Luciferase Assay System from Promega (catalog #: E2610), with luminescence signals measured using the Gen5 software of a Cytation 5 Cell Imaging Multi-Mode Reader (BioTek, Winooski, VT, USA).

**Immunofluorescence staining.** HeLa cells and HEK293T cells were used for transfection. Cells were seeded on 35-mm glass bottom dishes and transfected with LOCa3-FLAG. 24 h after transfection, cells were stained with 4% PFA for 20 min at room temperature. After fixation, cells were blocked by 3% BSA for 1 h. Then, an anti-FLAG monoclonal antibody (Sigma-Aldrich, # F3165, 1:300) was added to glass dish and incubated at 4 degrees overnight. An anti-mouse Alexa Fluor-488 IgG (ThermoFisher Scientific # A-11029) was diluted at 1:1000 to label the primary FLAG antibody at room temperature for 1 additional hour. PBS washing was repeated 3 times for each step to minimize unspecific binding.

**Cell viability assay.** To monitor light-induced NFAT-dependent cell death, NFAT RE-MLKL-NT and LOCa3 (or LOCa3-E106A as negative control) were co-transfected into HEK293T cells seeded onto glass-bottom dishes. Sixteen hours after transfection, cells were exposed to pulsed blue light with (1 min ON + 9 min OFF for 16 h) in the presence of co-stimulatory PMA (15 nM). Cells treated with the same concentration of PMA were kept in the dark as a control. Cells were treated with the SYTOX blue dye (Thermo Fisher Scientific, S11348, 1:5000 dilution, $C_f = 1$ μM) before imaging. Images were acquired using a 10x objective (8 fields for each condition).

**RNA isolation and quantitative real-time PCR.** HEK293T cells were seeded in 24-well plates and then transfected with LOCa3, sgRNAs (targeting *MYOD1*) and the CaRROT[35] plasmids. Twenty-four hours post-transfection, samples were stimulated with blue light (470 nm, 40 μW/mm$^2$, cycles of 1 min ON + 9 min OFF) or kept in the dark as a control. Total RNA was extracted using TRIzol (Invitrogen, Carlsbad, CA, USA) and then reverse-transcribed using amfiRivert Platinum One cDNA Synthesis Master Mix (GenDEPOT, Barker, TX, USA). Relative expression levels were determined using an SYBR Green real-time PCR kit (GenDEPOT, Barker, TX, USA) and data were normalized to the *GAPDH* mRNA level with the delta-delta Ct method.

**Electrophysiological measurements.** The light-induced current in HEK293 cells expressing YFP-LOCa3 was measured with a conventional whole-cell recording method using a HEKA EPC 10 USB double patch amplifier as we routinely use[53]. The pipette solution contained (mM): 135 Cs-aspartate, 8 MgCl$_2$, 10 EGTA, and 10 Cs-HEPES (pH 7.2). The extracellular solution contained (mM): 130 NaCl, 4.5 KCl, 20 CaCl$_2$, 10 TEA-Cl, 10 d-glucose, and 5 Na-HEPES (pH 7.4). A 10-mV junction potential was applied to compensate for the liquid junction potential. For characterization of the Ca$^{2+}$/Na$^+$ selectivity of LOCa3, HEK293 cells (ORAI-knockout; or OK) transfected with mKate2-P2A-LOCa3 were bathed in a 0 Ca$^{2+}$ + 130 mM Na$^+$ extracellular solution, following by whole-cell current recording upon light stimulation.

**HSPC isolation and retroviral transduction.** Retrovirus was packaged in plat-E cells by co-transfection of the MSCV-IRES-mCherry-based vector and the PCL-ECO helper plasmid[54]. $4 \times 10^7$ bone marrow cells were harvested from the femurs of WT or *Tet2$^{-/-}$* C57BL/6 mice. After red blood cell lysis, the bone marrow cells were stained with a panel of biotin-conjugated antibodies (1:10 dilution). Specifically, antibodies against Ter-119 (BioLegend Ter119, 1:50), Gr1 (BioLegend RB6-8C5, 1:50), Mac-1 (BioLegend M1/70, 1:50), B220 (BioLegend RA3-6B2, 1:50), and CD3 (BioLegend 17A2, 1:50) surface antigen were mixed as a cocktail for lineage

labeling. The cells were washed once and then incubated with anti-biotin microBeads (cat number: 130-090-485) following the manufacturer's instructions, followed by negative selection using the MACS LS Columns (cat number: 130-042-401). The flow-through was considered as the Lineage negative fraction, also known as HSPCs. The Lin− cells were cultured in StemSpan SFEM (StemCell Technologies) supplemented with 10% FBS and recombinant murine IL-3 (10 ng/ml) (PeproTech, Catalog Number:213-13), recombinant murine IL-6 (10 ng/ml) (PeproTech Catalog Number:216-16), and recombinant murine SCF (50 ng/ml) (PeproTech, Catalog Number:250-03). Twenty-four hours later, WT and *Tet2$^{-/-}$* HSPC cells were infected with a retrovirus expressing LOCa3. Cells were infected twice with 24 h in between to boost the transduction efficiency. The FlowJo (v10.5.3) software was used for all flow cytometry analysis.

For ex vivo culture, LOCa3-expressing cells were plated at $2 \times 10^5$/ml and grown for 3–5 days in StemSpan SFEM (StemCell Technologies) supplemented with recombinant murine IL-3 (10 ng/ml) (PeproTech, Catalog Number: 213-13), recombinant murine IL-6 (PeproTech Catalog Number:216-16), and recombinant murine SCF (50 ng/ml) (PeproTech, Catalog Number:250-03). Cells were kept in the dark or stimulated with blue light for 30 min every day.

For analysis of the bone marrow hematopoietic progenitor stem cells (HSPCs) marked by LSK (Lin-c-Kit+Sca-1+), cells were incubated with a panel of biotin-conjugated antibodies for lineage labeling followed by secondary streptavidin-eFluor®450 (ThermoFisher Scientific 48-4317-82, 1:400) staining, c-Kit APC (ThermoFisher Scientific 2B8, 1:200), Sca-1 FITC (ThermoFisher Scientific D7, 1:200)

**Fly stocks and rearing.** W1118 WT and elav-Gal4; cyo/UAS-Aβ$_{42}$ *Drosophila* strains were gifts from Dr. Zhou Bing's Lab at Tsinghua University. UAS-LOCa3 (attP40) and UAS-jRCaMP1b (attP2) flies were generated by the Tsinghua Fly Center[55]. These strains were then crossed to generate the corresponding AD (Aβ$_{42}$) flies co-expressing LOCa3 with or without jRCaMP1b. Tissue-specific expression of target genes was achieved by using the pan-neuronal elav-GAL4c155 driver[56]. All strains were grown in standard corn flour agar media at 25 °C.

**Climbing assay.** The climbing assay was carried out within a plastic vial (2 cm diameter) similar to those described earlier[57]. Ten flies of the appropriate genotype were placed in the vial, which was gently tapped on its bottom. The startle-induced geotactic response or the climbing ability was monitored by counting the number of flies traveling across a distance of 10 cm within 10 s. Five different batches of flies were tested for each time point, with the results plotted using the Prism 8 software (GraphPad) as mean ± s.e.m.

**Ca$^{2+}$ imaging in the *Drosophila* brain.** *Drosophila* brains were dissected out, immersed in Ca$^{2+}$ imaging solution containing 5 mM CaCl$_2$ (PH 7.4)[53], and immobilized to the bottom of 35 mm dishes with a water-polymerizing surgical glue[58,59]. jRCaMP1b fluorescence was acquired using an LSM 880 NLO confocal microscope (Zeiss) using a 20X water-immersion lens. To measure the change of Ca$^{2+}$ level induced by blue light, time-lapse images were collected every 4 s for 5 min. After 20 images were acquired as a baseline, an external blue light source (470 nm, 40 μW/mm$^2$) was applied to stimulate the brain. The fluorescence readings from regions of interest were exported from the software and imported into MATLAB 2014a and plotted with GraphPad Prism software.

**Statistics and reproducibility.** All the data were plotted and shown as mean ± s.e.m. (unless otherwise noted) by using the GraphPad Prism 8.3.0 graphing software. The analyzed number (*n*) of samples are described in the figure legends for each experiment. The half-lives were also determined by using the GraphPad Prism software package. All the experiments were carried out with at least three independent biological replicates unless otherwise noted. Statistical analysis was performed using unpaired Student's *t*-test. *$P < 0.05$; **$P < 0.01$; ***$P < 0.001$; ****$P < 0.0001$.

**Reporting Summary.** Further information on research design is available in the Nature Research Reporting Summary linked to this article.

## Data availability
Supplementary Data are available online. The plasmids and all other data are available from the corresponding author upon reasonable request. Source data are provided with this paper.

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

## Acknowledgements

This work was supported by the Ministry of Science and Technology of China (2019YFA0802104 to Y.W.), the National Natural Science Foundation of China (91954205 and 31671492 to Y.W.), the Welch Foundation (BE-1913-20190330 to Y.Z.), the American Cancer Society (RSG-16-215-01-TBE to Y.Z. and RSG-18-043-01-LIB to Y. H.), and the John S. Dunn Foundation (to Y.Z.). We would like to thank the Experimental Technology Center for Life sciences, Beijing Normal University, for the support on in vivo studies. We thank Dr. Reid Powell at the Institute of Biosciences and Technology, Texas A&M University, for his help in analyzing the high-throughput screening data.

## Author contributions

Y.Z., Y.H., and Y.W. conceived the ideas, designed the study, and directed the work. L.H. designed and generated all the plasmid constructs. L.H., L.W., P.T., and G.M. developed and characterized light-induced Ca$^{2+}$ influx. L.H. performed the optogenetic control of luciferase expression, cell viability and gene expression experiments. H.Z. isolated HSCs from mice and performed HSC-related experiments. L.W. and Y.L. generated the

transgenic fly and did fly related experiments with help from L.S. and F.D.; L.H., H.Z., and L.W. analyzed data. S.Z. performed the whole-cell current recordings. L.S. and F.D. provided intellectual input. Y.Z., Y.W., Y.H., L.H., S.S., and P.T. wrote the manuscript.

## Competing interests

Y.Z., L.H. and Y.H. have submitted a patent application to the United States Patent and Trademark Office pertaining to the design and biomedical applications aspect(s) of this work (application number 63/127,506). The remaining authors declare no competing interests.
