## [Peer Review File · Nature Communications]

Reviewers' Comments:

Reviewer #1:

Remarks to the Author:

The authors present a fascinating account of studies to develop and use a light operated Orai calcium channel and show its operation and utility to examine and understand calcium signaling events. It is an excellent paper and of broad significance to a wide range of investigators. The tool developed here is powerful and definitely has great advances over previous work to develop light-operated calcium signaling modifiers. The authors cleverly screened and developed constructs, and the experiments are very well designed with many important controls are included.

The paper has high impact by developing a new optogenetic tool that is highly specific for inducing physiologically relevant changes in Ca²⁺ homeostasis. The authors have clearly described how they designed this light-operated Ca²⁺ channel (LOCa) using a constitutively active Orai1 mutant – Orai1-P245T as a platform. The photosensory module, LOV2, was inserted into the TM2-TM3 loop of the Orai1-P245T mutant between residues S163 and P164. Using mutagenesis and high-throughput screening, the authors identified a final version of the light-activated Ca channel (LOCa3), Orai1-LOV2-H171D-P245T. This light-regulated Orai1 channel preserves its high Ca²⁺ selectivity and classical CRAC current properties, and completely emulates the effect of STIM1-mediated Orai1 gating and downstream effects. Perhaps most importantly, the authors have also demonstrated the use of this new optogenetic tool in regulating Ca²⁺-dependent functions at both the cellular and animal level.

The data indicate that LOCa3 is capable of modifying Ca²⁺ signaling in an important hematopoietic stem cell system, precisely controlling gene expression and cell function. The authors also investigated the feasibility of using LOCa3 for light-switchable transcriptional programming, specifically in mediating neurodegenerative changes in vivo. This lab had previously developed a STIM1-based optogenetic tool that could sequentially open Orai1 channels. However, this tool had limitations since STIM proteins have targets other than just Orai channels. As the authors emphasize, the Orai1 channel has the highest Ca²⁺ selectivity. Thus, the new optogenetic tool in which the Orai1 channel itself is activated directly by blue light has very broad development prospects, not only in stem cells and neurodegenerative studies, but perhaps also in solid tumor therapy due to the utility of precise spatial control. Significantly, the total size of the construct (less than 1.5kb) allows it to be easily packaged in a variety of viral vector systems with broad potential diagnostic and therapeutic applications. Therefore the innovative impact of this tool is impressive.

The authors have been exceedingly careful and comprehensive in their studies. The results are extremely interesting and provide a valuable tool which will affect many fields. The methods used are clear and well described, the experiments have been carefully carried-out and the conclusions drawn are convincing. Hence, it is difficult to suggest further experiments to support the authors' conclusions. Nevertheless, there are several suggestions to consider for some minor changes to clarify and optimize the paper.

1. Page 4, line10, for LOCa1, where was the LOV2 inserted? Same as LOCa2?
2. Does the LOCa3 have a dominant negative effect? Normally, Orai channels over-expressed in cells will have quite severe effects to dominate and reduce endogenous SOCE? This is very important for potential applications. The authors need to test and reveal its effects on endogenous SOCE.
3. The cartoon in Fig. 1a is not accurate and is misleading. Of course the Ca²⁺ ions do not pass through TM2/TM3. The authors need to modify this cartoon. The pore is of course formed by TM1.
4. Page 8, line 19 "by contrast, the control group did...". Please state clearly what exactly the

control group is. Is it LOCa3-E106A or something else?

5. Please give a full name when the abbreviation when first used. For example, in page 3, line 13, ChR2 is first used here, please write channel rhodopsin before it. And in the abstract, please add the full name for LOV2, and other abbreviations where necessary.

Reviewer #2:

Remarks to the Author:

He et al. developed a light-gated Ca²⁺ channel comprising an engineered ORAI1 and the blue light-sensitive LOV2 domain through impressive engineering efforts. The author then demonstrated a few applications of this light-controlled Ca²⁺ channel, which they name LOCa, for example to suppress the self-renewal of Tet2 deficient stem cells, and control gene transcription by light by combining LOCa with their previously developed CaRROT system. The authors also applied the LOCa system in a Drosophila model and induced Ca²⁺ influx in vivo.

This work describes a novel light-inducible Ca²⁺ channel and several interesting applications. The conclusions are generally supported by the data. There are only a few questions that need to be addressed before publication:

1. The specificity of the synthetic Ca²⁺ channel should be carefully addressed. In Figure 1j, both Ca²⁺ and Sr²⁺ influxes were monitored by a Ca²⁺ specific indicator, jRCaMP1b, which is not known to respond to Sr²⁺ since a similar calmodulin and M13-based Ca²⁺ biosensor, G-CaMP, has a 1000-fold higher K_d towards Sr²⁺ than to Ca²⁺ [Ohkura, M., et al. *Anal. Chem.* 77, 5861-5869 (2005)]. Moreover, as the author mentioned in their introduction, a key feature of the store-operated calcium channel is its Ca²⁺/K⁺ selectivity. Although ORAI1 itself is selective for Ca²⁺, the Ca²⁺/K⁺ selectivity of the highly engineered channel needs verification.

2. The rationale of the in vivo experiment is not clearly presented. The demonstration that LOCa works in vivo is very helpful. However, the sentence "we envision that optogenetic stimulation of neuronal calcium signaling might rescue the impaired neuronal SOCE activity to alleviate neurodegeneration" may generate some confusion. Are the authors proposing to use optogenetic stimulation as a therapeutic strategy?

3. The authors should discuss the limitation of LOCa as a new optogenetic tool. For example, the kinetics of Ca²⁺ flux induced by LOCa are slow compared to channelrhodopsin therefore LOCa might not be suitable for acute stimulation of neuronal activities. It is important to provide the audience with a complete picture.

Reviewer #3:

Remarks to the Author:

In this manuscript, Zhou and co-authors reported the development of a series of engineered light-activatable Ca²⁺ channels, LOCa, by direct insertion of LOV2 domain into ORAI1 Ca²⁺ channel. The authors characterized and demonstrated the utilities of LOCa in live cells and in vivo in Drosophila. These light-activatable Ca²⁺ channels are innovative within this class, representing a valuable addition to the optogenetics toolbox for controlling Ca²⁺ dynamics in cells. LOCa will likely be widely adopted by the research communities due to its simple yet effective design. Overall, I am enthusiastic about the new tool design and solid characterization work. Thus I think this manuscript is potentially suitable for publication in Nature Communications. Yet I would like to raise a few technical issues for authors to address to possibly improve upon the current manuscript and to better inform and serve the user community. The following are my suggestions and comments:

The wildtype ORAI1 has exceptional Ca²⁺ selectivity, however, whether this engineered ORAI with LOV2 inserted retains such ion selectivity remains elusive to me. It would be crucial to characterize and verify the selectivity of LOCa.

Engineered channels might have low membrane expression/localization due to impaired trafficking, leading to relatively poor performance. It would be appropriate for the authors to include detailed images of LOCa surface expression and comment on its cellular localization.

Light sensitivity of optogenetics tools is an essential parameter for end-users as higher photosensitivity reduces photodamage from intensive light illumination. Most of the activation light intensity used in this work is 40 $\mu\text{W}/\text{mm}^2$, it will be better if the authors could include a characterization of LOCa performance at varying light densities.

It would be appropriate for the author to have a more systematic and comprehensive comparison between LOCa with other existing optogenetic Ca²⁺-modulating approaches (such as Opto-CRAC, optoSTIM, monSTIM1, and BACCS), especially in terms of kinetics, light sensitivity, etc., to help the potential users to choose the right tool for their applications. Direct experimental comparison would be ideal, a detailed discussion could be sufficient as well.

The engineering approach of attaching the LOV2 domain to an ion channel is conceptually similar to the light activatable potassium channel BLINK1. It would be appropriate to discuss BLINK1 engineering and cite the original paper.

The authors stated "LOV2 was initially inserted into different regions...including the N/C termini and...", however, the C terminal result is not shown in Supplementary Fig. 1a.

It's great the authors tested the basal Ca²⁺ concentration with the expression of LOCa (Supplementary Fig. 2b), but the expression level of LOCa3 is not clear in the case. It would be better to include various expression levels (possibly with varying DNA amounts in transfection).

In the experiment described in Supplementary Fig. 1b, please introduce a proper control to eliminate the possible artifact from R-GECO1.2 photoactivation, or use jRCaMP1b instead.

For Fig. 1f and 1i, please label the construct name on the panels to avoid confusion. The same applies to Fig. 1k, and 1m.

Author affiliation #1 and #3 are exactly the same.

Editorial comments:

"Thank you again for submitting your manuscript "Engineering of a bona fide light-operated calcium channel" to Nature Communications. We have now received reports from 3 reviewers and, on the basis of their comments, we have decided to invite a revision of your work for further consideration in our journal. Your revision should address all the points raised by our reviewers (see their reports below)."

Response: We thank the editor and all three reviewers for your important comments in regard to revising our manuscript. We have now thoroughly considered these important points and have conducted all the recommended experiments. We have also made clarifications to address the questions raised by all three Reviewers. All changes made in the main text and supplementary materials were highlighted in blue.

The point-to-point responses to the comments are included as follows.

Reviewer #1

We are truly thankful to the reviewer for the supportive remark that *"The authors have been exceedingly careful and comprehensive in their studies. The results are extremely interesting and provide a valuable tool which will affect many fields"*

Minor concerns:

1.1. *"Page 4, line10, for LOCa1, where was the LOV2 inserted? Same as LOCa2?"*

Response: We thank the reviewer for mentioning this. In LOCa1, LOV2 was inserted between residues R167 and M168 in the intracellular loop (S8 site in **Fig. 1c**). In LOCa2, we optimized the LOV2 insertion site at position S6 between residues S163 and P164, which led to enhanced light-induced Ca²⁺ changes (**Fig. 1d**). We have revised the related text and figure legends to better explain these engineering efforts.

1.2. *"Does the LOCa3 have a dominant negative effect? Normally, Orai channels over-expressed in cells will have quite severe effects to dominate and reduce endogenous SOCE? This is very important for potential applications. The authors need to test and reveal its effects on endogenous SOCE."*

Response: We greatly thank the reviewer for this valuable suggestion. We have followed the reviewer's advice and examined endogenous SOCE in HEK293 cells, with and without co-expression of LOCa3. Cells were pre-incubated with 1 μ M TG for 10 min to ensure full store depletion, and a red Ca²⁺ indicator, jRGECO1a, was used to monitor the SOCE responses. It turned out that HEK293 cells expressing LOCa3 showed slightly attenuated SOCE responses (**Fig. A1**, new **Supplementary**

Fig. 3c). Thus, LOCa3 might exert a limited dominant negative effect on endogenous SOCE (rather than a severe one). However, as pointed out by the reviewer, this effect might be mitigated by lowering the expression level of the engineered ORAI channel.

Figure A1: Effects of LOCa3 on endogenous SOCE response in HET293 cells. To deplete ER Ca^{2+} store, cells were kept in nominally Ca^{2+} free imaging solution containing $1 \mu\text{M}$ thapsigargin (TG) for 10 min. TG was present throughout the recordings. (n=3).

1.3. “The cartoon in Fig. 1a is not accurate and is misleading. Of course the Ca^{2+} ions do not pass through TM2/TM3. The authors need to modify this cartoon. The pore is of course formed by TM1.”

Response: We deeply appreciate the reviewer’s constructive suggestion. We have followed the reviewer’s suggestion and modified the cartoon as follows (**Fig. A2**, or revised **Fig. 1a**). Since TM1 constitutes the Ca^{2+} ion conduction pathway, an arrow was used to indicate Ca^{2+} influx.

Figure A2: Schematic depiction of photoswitchable Ca^{2+} influx through an engineered ORAI1 Ca^{2+} channel. The LOV2 domain is inserted into the intracellular loop of a constitutively-active ORAI1 (caORAI1) and maintains the hybrid channel in a closed state in the dark. Upon light illumination at 470 nm, conformational changes within LOV2 trigger allosteric activation of engineered ORAI1 to evoke Ca^{2+} flux across the plasma membrane.

1.4. "Page 8, line 19 "by contrast, the control group did....". Please state clearly what exactly the control group is. Is it LOCa3-E106A or something else?"

Response: The control group used herein refers to AD flies without LOCa3 co-expression. One sentence has been added into the revised manuscript to reflect this fact.

"By contrast, the control group (AD flies without LOCa3 co-expression) did not exhibit light-induced changes in the jRCaMP1b fluorescence (Fig. 3c)."

1.5. "Please give a full name when the abbreviation when first used. For example, in page 3, line 13, ChR2 is first used here, please write channel rhodopsin before it. And in the abstract, please add the full name for LOV2, and other abbreviations where necessary."

Response: We have followed the reviewer's suggestion to add the full name for the abbreviation "channelrhodopsin-2 (ChR2)" when first used.

Reviewer #2:

We deeply appreciate the Reviewer's positive remarks in regard to the innovation in this work, and for your valuable comments and suggestions. As described below, we have performed the recommended experiments and added sufficient discussions to address all major concerns.

2.1. "The specificity of the synthetic Ca^{2+} channel should be carefully addressed. In Figure 1j, both Ca^{2+} and Sr^{2+} influxes were monitored by a Ca^{2+} specific indicator, jRCaMP1b, which is not known to respond to Sr^{2+} since a similar calmodulin and M13-based Ca^{2+} biosensor, G-CaMP, has a 1000-fold higher K_d towards Sr^{2+} than to Ca^{2+} [Ohkura, M., et al. *Anal. Chem.* 77, 5861-5869 (2005)]. Moreover, as the author mentioned in their introduction, a key feature of the store-operated calcium channel is its Ca^{2+}/K^+ selectivity. Although ORAI1 itself is selective for Ca^{2+} , the Ca^{2+}/K^+ selectivity of the highly engineered channel needs verification."

Response: We followed the reviewer's suggestion, and carried out more characterization on Ca^{2+} selectivity of LOCa and GECIs using HEK E74 cells that stably express L-type Ca^{2+} ($Ca_v1.2$) channels. We have performed the following experiments to address the specificity issue.

1) It is well-established that depolarization-induced $Ca_v1.2$ channel activation could mediate both Ca^{2+} and Sr^{2+} influxes. Indeed, when examined with jRCaMP1b, the depolarization-induced Ca^{2+} and Sr^{2+} responses through $Ca_v1.2$ channels were similar in amplitudes (Fig. A3a, or new **Supplementary Fig. 5a**). Furthermore, both depolarization-induced responses could be blocked by

nimodipine, an inhibitor of $\text{Ca}_v1.2$ channels. When examined with GEM-GECO, a more sensitive Ca^{2+} indicator with a larger dynamic range (~ 110 -fold ratiometric change in fluorescence; PMID: 21903779), the responses to Ca^{2+} and Sr^{2+} also remained comparable (**Fig. A3b**, or new **Supplementary Fig. 5b**). Even though GCaMP was reported to be not highly sensitive to Sr^{2+} in vitro (PMID: 16159115), our results indicate that at least jRCaMP1b and GEM-GECO exhibited similar sensitivity to Ca^{2+} and Sr^{2+} in HEK293 cells used in our study.

2) Since $\text{Ca}_v1.2$ mediated Ca^{2+} and Sr^{2+} responses equally well (**Fig A3a, b**, or new **Supplementary Fig. 5a-b**) but light-activated LOCa3 could only induce Ca^{2+} influxes but not Sr^{2+} responses (**Fig 1j**). We thus conclude that LOCa3 is a highly Ca^{2+} selective channel.

3) By “ $\text{Ca}^{2+}/\text{K}^+$ selectivity”, we assume that the reviewer actually meant “ $\text{Ca}^{2+}/\text{Na}^+$ selectivity” (Na^+ is the dominant cation in the extracellular media; ~ 130 mM Na^+ vs ~ 1 -2 mM Ca^{2+} in the culture media). We followed the reviewers’ advice and carried out whole-cell patch clamp recordings on LOCa3-expressing cells (HEK293 cells without ORAI or HEK-OK cells to avoid complications from endogenous ORAI-STIM signaling) bathed in 0 Ca^{2+} + 130 mM Na^+ extracellular solution. The results showed that LOCa3 channels failed to mediate any detectable light-activated Na^+ influx current (**Fig. A3c**, or new **Supplementary Fig. 5c**). This result strongly suggests that, similar to its prototype ORAI1 channels, engineered LOCa3 channel retains a high selectivity for Ca^{2+} over Na^+ .

Figure A3: Characterizations of Ca^{2+} selectivity of LOCa3 or $\text{Ca}_v1.2$ channels. Data were shown as mean \pm s.e.m. (a-b) In HEK-E74 cells transiently expressing jRCaMP1b (a) or GEM-GECO (b), high K^+ -induced Ca^{2+} (blue) or Sr^{2+} (black) influxes through $\text{Ca}_v1.2$ channels were examined. Left, typical traces, right, statistics. 2 μM nimodipine was used to block the activity of $\text{Ca}_v1.2$ channels. No significant difference between mean Ca^{2+} (blue) or Sr^{2+} (black) entry was found ($P > 0.1$, two-tailed Student’s t-test; $n=3$). (c) When bathed in nominally Ca^{2+} free solution, HEK OK (ORAI-knockout) cells expressing LOCa3 showed no detectable light-induced Na^+ current. *Left*, the mean time-course of whole-cell current recording; *Right*, the mean current-voltage relationship at the end of electrophysiological recording shown on the left ($n=5$).

2.2. *“The rationale of the in vivo experiment is not clearly presented. The demonstration that LOCa works in vivo is very helpful. However, the sentence “we envision that optogenetic stimulation of neuronal calcium signaling might rescue the impaired neuronal SOCE activity to alleviate neurodegeneration” may generate some confusion. Are the authors proposing to use optogenetic stimulation as a therapeutic strategy?”*

Response: We thank the reviewer for stating that *“The demonstration that LOCa works in vivo is very helpful”*. We apologize for not making our point clear enough. We have now rephrased the above-mentioned sentence as follows:

“We envision that optogenetic stimulation of neuronal calcium signaling might rescue the impaired neuronal SOCE activity, which has been linked to neurodegenerative disease in both animal models and patients (PMID: 30015245, PMID: 27881772).”

The goal here is to provide proof-of-concept evidence that optical stimulation of neuronal calcium signaling could be beneficial for alleviating symptoms seen in neurodegenerative models (as illustrated in a drosophila model of amyloid-beta toxicity in the study). The technology is still in a rudimentary stage and its potential as a therapeutic approach will be pursued in follow-on studies using rodent or even nonhuman primate models.

2.3. *“The authors should discuss the limitation of LOCa as a new optogenetic tool. For example, the kinetics of Ca²⁺ flux induced by LOCa are slow compared to channelrhodopsin therefore LOCa might not be suitable for acute stimulation of neuronal activities. It is important to provide the audience with a complete picture.”*

Response: We have followed the reviewer’s advice to provide the audience with a complete picture. A table (new **Supplementary Table 1**) listing the major parameters and pros/cons of major optogenetic tools for calcium signaling was added in the revised manuscript. For instance, as pointed out by the reviewer, acute stimulation of neuronal activities is indeed not suitable with LOCa3 or STIM1-based optogenetic tools given their on/off kinetics in the range of seconds or minutes (rather than milliseconds as seen in ChR2 or its variants). However, the relatively slow kinetics makes them more suitable for controlling other physiological processes, such as gene expression, T cell activation and cell metabolism.

Reviewer #3:

We greatly thank the Reviewer for the high enthusiasm on our study and the supportive remarks that *“LOCa will likely be widely adopted by the research communities due to its simple yet effective design. Overall, I am enthusiastic about the new tool design and solid characterization work. Thus I think this manuscript is potentially suitable for publication in Nature Communications.”*

3.1. *“The wildtype ORAI1 has exceptional Ca²⁺ selectivity, however, whether this engineered ORAI with LOV2 inserted retains such ion selectivity remains elusive to me. It would be crucial to characterize and verify the selectivity of LOCa.”*

Response: We thank the reviewer for this valuable suggestion. We performed more characterizations on Ca²⁺ selectivity of LOCa with different approaches. Please refer to our response to Comment 2.1 described above (see **Fig. A3**).

3.2. *“Engineered channels might have low membrane expression/localization due to impaired trafficking, leading to relatively poor performance. It would be appropriate for the authors to include detailed images of LOCa surface expression and comment on its cellular localization.”*

Response: We have followed the reviewer’s suggestion and performed immunofluorescence staining of LOCa3 expressed at least in two different cell types, HeLa and HEK293. To minimize the perturbation to LOCa3, we inserted a FLAG tag in the second extracellular loop that is known to tolerate epitope insertion without compromising its light-induced effect (**Fig. A4**, or new **Supplementary Fig. 2a**). When HEK or HeLa cells expressing LOCa3 tagged with an extracellular FLAG tag were stained with an anti-FLAG antibody and examined with confocal imaging, LOCa3-FLAG showed a typical PM-like distribution (**Fig. A4**, left panel). These results thus showed that LOCa3 had surface expression. We also attempted to tag fluorescence proteins (GFP, YFP or mCherry) to either the N- or C-terminus of LOCa3, but these constructs showed relatively poor PM localization with a significant portion trapped inside cells (mostly with ER-like distribution), a scenario that is often seen with FP-fused Ca_v1.2, TRPC, or potassium channels. Therefore, whenever possible, we mostly used a P2A self-cleaving peptide-based expression system (mKate-P2A-LOCa3) throughout this manuscript, which enables the co-expression of LOCa3 without any tag and a fluorescent protein as marker at a near 1:1 ratio (indicator for expression).

Figure A4: Localization of LOCa3. Confocal images of intact HeLa or HEK293T cells expressing LOCa3 with a FLAG tag inserted in the second extracellular loop between residues 206 and 207 with a GSGS linker on either side (left panel). Cells were left unpermeabilized and stained with an anti-FLAG antibody. Scale bar, 10 μm . The primary sequence of the LOCa3-FLAG junction region was shown above the confocal images. Right panel, FLAG tag insertion into LOCa3 did not affect light induced Ca^{2+} influx in HeLa cells. $n = 30$ cells.

3.3. “Light sensitivity of optogenetics tools is an essential parameter for end-users as higher photosensitivity reduces photodamage from intensive light illumination. Most of the activation light intensity used in this work is $40 \mu\text{W}/\text{mm}^2$, it will be better if the authors could include a characterization of LOCa performance at varying light densities.”

Response: We thank the reviewer for this constructive suggestion. We quantified the calcium responses in LOCa3-expressing HeLa cells, which were subjected to light illumination with increasing intensities ($1\text{-}40 \mu\text{W}/\text{mm}^2$; **Fig. A5**, or new **Supplementary Fig. 3b**). LOCa3-expressing cells showed a light intensity-dependent increase in Ca^{2+} influx, as reported by GCaMP6m signals.

Figure A5: Light-tunable Ca^{2+} entry in LOCa3-expressing HeLa cells. Maximal fold-changes of GCaMP6m signals were plotted against varying light power densities at 470 nm ($n = 25\text{-}31$ cells).

3.4. “It would be appropriate for the author to have a more systematic and comprehensive comparison between LOCa with other existing optogenetic Ca^{2+} -modulating approaches (such as Opto-CRAC, optoSTIM, monSTIM1, and BACCS), especially in terms of kinetics, light sensitivity, etc., to help the potential users to choose the right tool for their applications. Direct experimental comparison would be ideal, a detailed discussion could be sufficient as well.”

Response: We have followed the reviewer’s suggestion and added a table comparing LOCa3 with other optogenetic Ca^{2+} -modulating tools (new **Supplementary Table 1**).

3.5. “The engineering approaching of attaching the LOV2 domain to an ion Channel is conceptionally similar to the light activatable potassium channel BLINK1. It would be appropriate to discuss BLINK1 engineering and cite the original paper.”

Response: We have now cited the related paper and added the following sentence in the revised manuscript. “Inspired by successful engineering of a light-activatable viral potassium channel with two transmembrane regions (designated BLINK1) (PMID: 25954011), we set our eyes on a more challenging target, the ORAI1 channel made of four-pass transmembrane domains and assembled as a hexamer in the plasma membrane (PMID: 23180775, PMID: 30160233).”

3.6. “The authors stated “LOV2 was initially inserted into different regions...including the N/C termini and...”, however, the C terminal result is not shown in Supplementary Fig. 1a.”

Response: We thank the Reviewer for bringing this point to our attention. We have added the data for C terminal insertion in **Fig. A6**, or revised **Supplementary Fig. 1a**. This construct failed to evoke light-induced intracellular Ca^{2+} changes.

Figure A6: Strategies employed to design photoswitchable ORAI1 Ca^{2+} channels. GCaMP6m signals before and after blue light illumination were plotted as bar graphs below the cartoons. n=16-40 cells.

3.7. "It's great the authors tested the basal Ca^{2+} concentration with the expression of LOCa3 (Supplementary Fig. 2b), but the expression level of LOCa3 is not clear in the case. It would better to include various expression levels (possibly with varying DNA amounts in transfection)."

Response: We greatly thank the reviewer for this suggestion. We performed more repeats, and added a plot showing the correlations between the expression levels of LOCa3 and basal Ca^{2+} levels indicated by Fluo-4 in each individual cell (a total of 105 cells). No obvious positive/negative correlation was found between Ca^{2+} levels indicated by Fluo-4 and LOCa3 expression levels reflected by mKate2 intensity ($R^2=0.1$; Fig. A7, or new Supplementary Fig. 3a, left panel). Furthermore, in the dark, mKate2-P2A-LOCa3 transfected cells showed similar Fluo-4 background signals as non-transfected cells, further confirming minimized background activation and lack of concentration dependence.

Figure A7: Quantification of light induced Ca^{2+} changes in HeLa cells expressing mKate2-P2A-LOCa3 at varying degrees (indicated by mKate2 fluorescence intensities). The averaged Fluo-4 background signal (mKate2 negative cells) was indicated by the green line. The Fluo-4 signals for mKate2 positive cells at dark and light states were plotted in the left panel. Right panel, light-induced Fluo-4 intensity changes. No significant difference was noted between LOCa3 transfected or non-transfected cells ($n=77-105$ cells).

3.8. "In the experiment described in Supplementary Fig. 1b, please introduce a proper control to eliminate the possible artifact from R-GECO1.2 photoactivation, or use jRCaMP1b instead."

Response: We deeply appreciate the reviewer's suggestion. We have followed the advice and added the blue light-stimulated response curve for R-GECO1.2 to show the false-positive artifacts (Fig. A8, or revised Supplementary Fig. 1b, right panel). The results showed that artificial increase of R-GECO1.2 signals was discerned (right panel), but at a level that was substantially lower than signals arising from LOCa3 activation under the same illumination condition (left panel). In addition, we monitored light-induced Ca^{2+} response using jRaMP1b as suggested by the reviewer (Fig. 1j). Please note that we used a variety of calcium indicators (including GCaMP6m, R-GECO1.2,

jRCaMP1b, GEM-GECO, and Fluo-4) in the study, and all of them showed robust light-induced Ca^{2+} influx in LOCa3-expressing cells.

Figure A8: Light-inducible Ca^{2+} flux reported by red-emitting calcium indicator. Reversible control of Ca^{2+} signals in HEK293 cells. LOCa3-expressing cells were exposed to three repeated light-dark cycles, with the intracellular Ca^{2+} changes monitored by R-GECO1.2 (left panel). The artifacts for R-GECO1.2 caused by blue light were shown (right panel). n=10 cells.

3.9. “For Fig. 1f and 1i, please label the construct name on the panels to avoid confusion. The same applies to Fig. 1k, and 1m.”

Response: We have added the label “LOCa3” in the related panels (**Fig. 1f, 1i, 1k and 1m**).

3.10. “Author affiliation #1 and #3 are exactly the same.”

Response: We thank the reviewer for pointing out this mistake. We have now corrected the affiliation.

Reviewers' Comments:

Reviewer #1:

Remarks to the Author:

No further comments for the authors

Reviewer #2:

Remarks to the Author:

The reviewers' comments have been adequately addressed.

Reviewer #3:

Remarks to the Author:

The authors have done a great job addressing all my comments and questions. I would like to recommend the revised manuscript for publication in Nature Communications.